# Can a Forest Tree Species Progeny Trial Serve as an Ex Situ Collection? A Case Study on *Alnus glutinosa*

**DOI:** 10.3390/plants12233986

**Published:** 2023-11-27

**Authors:** Rita Verbylaitė, Filippos A. Aravanopoulos, Virgilijus Baliuckas, Aušra Juškauskaitė, Dalibor Ballian

**Affiliations:** 1Institute of Forestry, Lithuanian Research Centre for Agriculture and Forestry, LT-58344 Kėdainiai District, Lithuania; aravanop@for.auth.gr (F.A.A.); virgilijus.baliuckas@lammc.lt (V.B.); ausra.juskauskaite@lammc.lt (A.J.); 2Faculty of Agriculture, Forestry and Natural Environment, Aristotle University of Thessaloniki, P.O. Box 238, GR 541 24 Thessaloniki, Greece; 3Faculty of Forestry, University of Sarajevo, Zagrebacka 20, 71000 Sarajevo, Bosnia and Herzegovina; balliandalibor9@gmail.com; 4Slovenian Forestry Institute, Večna pot 2, 1000 Ljubljana, Slovenia; 5Academy of Sciences and Arts of Bosnia and Hercegovina, Bistrik 7, 71000 Sarajevo, Bosnia and Herzegovina

**Keywords:** black alder, genetic diversity, allele pattern, microsatellite genotypes, ex situ collection

## Abstract

Scientifically informed decisions for the long-term conservation of extant genetic diversity should combine in situ and ex situ conservation methods. The aim of the present study was to assess if a progeny plantation consisting of several open pollinated (OP) families and established for breeding purposes can also serve as an ex situ conservation plantation, using the case study of a Lithuanian progeny trial of *Alnus glutinosa*, a keystone species of riparian ecosystems that warrants priority conservation actions. We employed 17 nuclear microsatellite (Simple Sequence Repeat) markers and compared the genetic diversity and copy number of the captured alleles of 22 OP progeny families from this plantation, with 10 wild *A. glutinosa* populations, originating from the two provenance regions of the species in Lithuania. We conclude that the progeny plantation could be used as an ex situ plantation for the *A. glutinosa* populations from the first provenance region (represented by eight genetic conservation units (GCU)). Based on the present study’s results, we can expect that the *A. glutinosa* progeny plantation harbors enough genetic diversity of wild *A. glutinosa* populations from the first provenance region. This progeny plantation can serve as a robust ex situ collection containing local alleles present in at least one wild population with at least 0.05 frequency with 25 replications.

## 1. Introduction

Black alder (*Alnus glutinosa* (L.) Gaertn.) is a species that is widespread in Europe, and which can be found in all climate zones from the Mediterranean to the European north [1]. As such, it can play a major role in climate change mitigation, especially in the management of river basins where it is (or can become) an irreplaceable species [2,3]. In addition to its role in watershed management, black alder is very important in some European countries for biomass production, and breeding programs to increase productivity that include an intensive selection of plus trees are well under way [4].

The present rapid climate change and anthropogenic impact on ecosystems has caused an unprecedented species extinction rate [5,6]. Scientifically informed decisions are needed for the long-term conservation of extant species’ genetic diversity, combining in situ and ex situ conservation methods [7,8]. The Global Strategy for Plant Conservation (https://www.cbd.int/gspc/targets.shtml, accessed 20 August 2023) committed to preserve 70% of genetic diversity of crops, their wild relatives, and other economically or culturally valued species by 2020, but this goal has not been achieved. The amount of genetic diversity currently safeguarded in ex situ collections varies considerably species-wise. Most ex situ collections do not sufficiently represent the genetic diversity of wild populations [9]. The number of genotypes conserved in ex situ collections and their origin largely depends on the taxon being conserved. An ideal collection sample size can be inferred by empirical research at the species level [10].

The debate on the appropriate number of seeds or plants that should be kept in the ex situ collections to preserve species’ genetic diversity and evolutionary potential started almost 50 years ago [11]. The first quantitative estimate was 30–60 individuals, when the goal was to capture non-rare alleles [11]. Approximately N = 50 individuals per population was used as a “rule of thumb” by many ex situ collection curators and collectors [12]. However, the appropriate number of individuals to be sampled often differs from N = 50 and depends on aspects such as population size, reproductive biology, recent population history, and connectivity among populations [12,13,14,15,16,17]. Two sampling strategies represent extremes of possible population genetic subdivision: (a) Brown and Marshall [18] recommend collecting N = 50 individuals from each population, assuming all populations are unique genetically; and (b) Brown and Briggs [19] advocate that sampling N = 50 individuals is sufficient for a whole species, if sampled individuals are distributed among all populations, assuming that all populations are genetically homogenous. When applying theory to practice, the best strategy lies somewhere in between [20]. Computer simulations reveal that the number of populations and population sizes are two of the most important factors affecting the number of samples needed to represent genetic diversity present in wild populations [20]. The number of individuals chosen for an ex situ collection, depending on the targeted allele frequencies one wishes to capture within an ex situ collection (from low-frequency alleles (0.01–0.10) to local alleles (>0.05) to all alleles), varies from 30 to 800 individuals (Table 1) [20].

The other aspect to consider for an appropriate ex situ collection is the number of allelic copies it contains. Alleles represented only once in the collection have a high chance of being lost [21]. A secure collection should have at least five copies of target frequency alleles, and a robust collection at least 25 copies [20,22]. Capturing the desired number of copies of each targeted allele evidently leads to an increased sampling effort (Table 1) [20].

The majority of the economically important tree species in Europe have wide distribution ranges [23]. For many of these tree species there are clonal collections, seed orchards, and progeny/provenance trials established for breeding. To establish and maintain those plantations, governments and private companies invest considerable amounts of funding. Good and optimal practice could call for (at least) some of these plantations also being used for conservation purposes. In this respect, special attention should be given to older forest tree trials that are already of seed-bearing age and, in case of an extreme devastating event, they could be used in reforestation efforts, even in the restoration of existing in situ conservation units. However, to use established forest tree plantations as ex situ conservation plantations, we must determine if the genetic material present in the plantations sufficiently represents the genetic diversity of natural wild populations.

*A. glutinosa* (L.) Gaertn. (black alder) is an important, monoecious, self-incompatible, wind-pollinated keystone tree species found in riparian ecosystems in Europe [24,25]. It is known for its nitrogen fixing abilities [26], harbors extensive genetic diversity [27,28,29,30,31,32], and its populations are locally adapted [27,33]. Riparian floodplain forests are listed as priority conservation habitats in European Union countries under Annex I of the European Habitats Directive (92/43/EEC) and are under threat due to changing climate and anthropogenic impact [34]. *A. glutinosa,* as a keystone species of riparian ecosystems, warrants priority conservation efforts. To preserve the local genetic diversity of *A. glutinosa* populations, ex situ conservation should also be considered in addition to in situ conservation. It has been suggested that ex situ conservation of *A. glutinosa* should incorporate a large enough number of genetically different individuals (seeds sampled from 200 to 300 trees if it is a progeny collection, and from at least 100 trees if it is a clonal collection, will carry enough diversity) to maintain a viable breeding population that could be used for conservation as well as breeding purposes [35].

The aim of the present study was to assess if an existing progeny trial established for breeding purposes can also serve as an ex situ conservation collection, using a case study of an *A. glutinosa* progeny plantation from Lithuania.

## 2. Results

For the genetic analysis of *A. glutinosa* progeny plantation samples, we retained 17 loci. Three loci (Ag14, Ag20 and Ag23) were excluded, as they failed to amplify in 81, 64, and 70 of the sampled individuals, respectively. All loci retained for further analysis revealed no presence of null alleles in ex situ samples (based on MicroChecker software [36] analysis results), while locus A7 showed 13% of null alleles in genetic conservation unit (GCU) samples.

The 17 loci used were polymorphic and amplified 7 (A2) to 21 (Ag35), and 7 (A2) to 23 (Ag35 and Ag25) alleles, in ex situ and GCU sample lots, respectively. The total number of alleles found in the progeny plantation was 232 (13.65 on average per locus), while in natural populations it was 272 (16.0 on average per locus). In the GCU sample lot 52 alleles were found that were absent in the ex situ sample lot, while 15 alleles found in the plantation were absent in the in situ samples (Appendix A). All studied trees were identified as unique genotypes.

Genetic diversity parameters for *A. glutinosa* sample lots (5 individual GCU populations; 5 GCU populations combined; 10 GCU populations combined; 6 individual OP families; six OP families combined; and 22 OP families combined) are given in Table 2 (genetic diversity parameters of all GCU populations and all OP families are given in Appendix A). All black alder OP families investigated were genetically diverse and showed high heterozygosity values. Observed heterozygosity was higher than expected for all OP families, and inbreeding coefficient values showed an excess of heterozygotes. Allelic richness values for all the OP families investigated were similar and only slightly lower than the effective number of alleles. The highest value for allelic richness and effective number of alleles was found in the Juodkrantė (JUO) population.

The genetic relationships of the different OP families and populations were revealed by principal coordinate analysis (PCoA) (Figure 1). The 2D PCoA graph depicted a clustering of the wild GCU populations in the center, while the ex situ OP families, which by and large originated from different wild *A. glutinosa* populations, presented a wide scatter peripheral to the wild populations cluster.

An important aspect in an effective ex situ conservation of genetic diversity is how well allelic diversity is represented in the ex situ collection. To evaluate this aspect in the progeny plantation, we divided the alleles of GCU populations into five categories [10]: all alleles, very common alleles (>0.10 frequency), common alleles (>0.05 frequency), low-frequency alleles (0.10–0.01), and rare alleles (<0.01 frequency). The numbers of alleles in natural GCU populations for different data sets and for different allele frequency categories, are presented in Table 3.

The number of alleles in different single wild populations was very similar in all categories, with common and low-frequency allele categories prevailing (Table 3). Allele frequencies differed when analyzing single populations, but when analyzing the bulk sample of 5 and 10 GCU populations, allelic categories remained stable and only a few alleles fell into different frequency categories (data not shown). Analyzing how many of the alleles found in a GCU population were also found in a single OP family originating from the same population (comparison 1, Figure 6) revealed that less than 50% of the total allele numbers observed in the former were found in the latter (Figure 2A–F). When the six OP families combined sample lot was evaluated, the percentage of alleles from single wild GCU populations found in the progeny plantation was on average 81.6% (comparison 2, Figure 6). When all the OP families combined sample lot was analyzed, the number of alleles from a single population found in the progeny plantation increased to 88.6% (comparison 3, Figure 6). When analyzing different allele frequency categories, the least-represented frequency category was rare alleles, which on average reached 77.8%.

When seeking to conserve species in ex situ collections, it is important to preserve as much genetic diversity as possible, representing not only single populations in a collection, but also multiple populations or (in widespread species case) the metapopulation. Therefore, in the present study, we not only analyzed single wild population alleles and their frequencies, but also those from the metapopulation. The number of alleles found in six OP progeny families originating from all mother tree populations (JUO, VIL, BAT, PAZ, MIK) was 75% of the total number of alleles found in the five natural populations (comparison 4, Figure 6), and when all 22 OP families were included, the total number of alleles found reached 212, or 85% of total allele number of the five natural populations (Figure 2G) (comparison 5, Figure 6). If the progeny plantation was used as an ex situ collection for all 10 GCU populations included in present study, 81% of alleles detected in GCU populations were present in this sample lot (Figure 2H) (comparison 6, Figure 6).

We also calculated the copy number of different category alleles found in the progeny plantation. The results are presented in Figure 3A–F. Figure 3A represents a situation wherein only alleles from one OP family were taken into account (comparison 1, Figure 6). Evidently, this number of related plants cannot represent the in situ population effectively. In this case, only 49% of all alleles were found, and if we analyzed the fraction of rare alleles this number dropped to 28%, of which only 6% of alleles were found in five or more copies. Six OP families (117 plants in total) (Figure 3B) (comparison 2, Figure 6) were enough to conserve 82% of alleles with at least one copy number, and 51% of alleles were present with more than five copies. This sample lot contained more than 70% of alleles found in single GCU populations, although if we take into account the rare alleles, only 64% were present. When the whole progeny plantation was included (Figure 3C) (comparison 3, Figure 6), the absent allele number dropped to 11%, and the copy number of alleles increased, with almost 50% of rare alleles present with more than five copies. If the progeny plantation represented five GCU populations (Figure 3D) (comparison 4, Figure 6), 75% of alleles would be found in the six OP families sample lot, and 46% of rare alleles would be missing. In Figure 3E,F (comparisons 5 and 6, respectively, Figure 6), the results regarding the allele frequencies of 5 and 10 GCUs in a 22-OP family sample are presented. The results were similar to those presented above, despite the different number of wild populations used. In total, 15% and 19% of alleles were missing in the 5- and 10-population sample lots, respectively. When considering the very common alleles, 95% of genetic diversity (alleles) was found in two OP families of the progeny plantation. Over 70% of common or very common alleles were present in the progeny plantation in 25 or more copies. When low-frequency alleles from 5- and 10-population sample lots were analyzed, close to 90% of alleles were present in the 22 OP family’s collection, and only 15% and 16% of alleles were found with less than a copy number of five. Rare alleles show a different pattern: 28% and 36% of missing alleles, and only 21% and 17% of rare alleles, were found with 25 or more copies in the 5- and 10-population sample lots, respectively. Approximately only one-third of rare alleles in both cases were found with five copies or more.

The expected copy number in the whole progeny plantation (all 76 OP families) is presented in Figure 4, where the best-fitted trendline shows the likely copy number for each category of allele. The trendlines are, in general, of high significance, as the R^2^ values of the equations reveals.

## 3. Discussion

The Vytėnai progeny plantation investigated in our study shows high genetic diversity. When compared with wild GCU populations [32], the genetic diversity found in the progeny plantation was even higher than the 10 GCU population average (*Ho* = 0.77 and 0.73, respectively), although this difference was not statistically significant. Expected heterozygosity was found to be higher in wild populations than in the progeny plantation. The progeny plantation and wild population’s inbreeding coefficients were close to zero (*F_IS_* = −0.01 and 0.04, respectively). The allelic richness of wild populations based on 16 diploid individuals on average was found to be 7.33 [32], while in the present study *Ar* was found to be 2.61 for ex situ OP families and 2.88 on average for wild populations. The low number of allelic richness might be explained by the low number of individuals it was calculated for (two diploid individuals). Genetic differentiation among wild populations was low [29,30,32,37]. The gene flow among wild populations was high, as can be inferred from the low population differentiation, a result concordant with the geographical proximity of natural stands. Black alder is a common species in Lithuania, and its stands cover approx. 163,800 ha, or 7.9%, of the total forest area [38]. Local black alder is thus considered to have a metapopulation structure. The metapopulation structure of *A. glutinosa* was confirmed by the PCoA analysis, where OP families originating from different wild populations scattered around the cluster of 10 GCU populations used for comparisons in the present study (Figure 1).

Analysis of the allele number found in single OP families from wild *A. glutinosa* populations agreed with data in the literature. Hoban et al. [10] suggest that 24–82 unrelated individuals are required to capture 95% of variation present in a population when a reduced data set is used, and where alleles with a frequency of below 0.005 are excluded from the calculations. The number of individuals analyzed in our study was lower in terms of unrelated individuals (as we used samples from OP families). When a single OP family with 20 individuals was analyzed, only the very common or common allelic diversity with a threshold of 70% was conserved. However, the progeny plantation contains many more OP families, and the fraction analyzed showed that when 22 OP families were considered, the conserved allele number increased almost up to 90% for all alleles. This number reaches the upper limit found for existing ex situ collections, while in the study of Hoban et al. [10] the captured genetic diversity varied from 28% to 91%. In this study, 26% of all the trees present in the plantation were analyzed, and it is safe to assume that, if all the trees were to be analyzed, the captured genetic diversity would be higher and would most likely reach the desired 95% threshold (Figure 4). The investigated progeny plantation after 30 years of establishment retains approx. 1600 *A. glutinosa* plants that belong to 76 OP families originating from the plus-trees of Lithuanian populations, so the sample number per population (taking into account the fact that some populations in the progeny plantation are represented by more than one OP family) varies from 4 to 130. Based on the sampling strategy in this study, we investigated samples from 20 different stands (populations) and 15–20 progeny trees per population (Appendix A).

An analysis of the captured allele copy number in the progeny plantation revealed that the plantation can be expected to safeguard most of the common and very common frequency alleles with 25 copies or more (83% and 82.5%, respectively), and most of the low-frequency alleles with at least 5 copies each (86%), regardless of the number of wild GCU populations involved (1, 5, or 10) (Figure 3C,E,F and Figure 4). Based on the allele copy number change in different allele frequency groups and the subsequent trendlines of change, it is highly likely that, in the progeny plantation as a whole, the captured genetic diversity presents 25 or more copies for very common to low-frequency alleles. As can be seen from the increase in allele copy number when analyzing 6 and 22 OP families, the proportion of low-frequency alleles represented by 25 copies or more increased more than three times (from 14% to 45%; Figure 3D,E and Figure 4). It is expected that 207 alleles from the total 272 found in 10 natural populations would be present in the progeny plantation with more than five copies.

The Vytėnai progeny plantation could be used as an ex situ collection for the *A. glutinosa* populations, representing the first provenance region of Lithuania (of two). Taking into account the much higher number of *A. glutinosa* plants present in the progeny plantation than in the investigated sample, it can reasonably be expected that this progeny plantation should be enough to encompass 25 replicates of local alleles present in at least one population with at least a 0.05 frequency, and five replicates of all alleles meeting the genetic diversity threshold of 95% [20].

Our case study suggests that common forest tree species progeny or provenance trials could be worthy candidates to serve as ex situ collections for economically important widespread forest trees. Progeny trials most often represent a rather limited species distribution area (one or several provenances from the same country), and in case of a severe but not widespread biotic outbreak or natural disaster, they could serve as a good source of genetic material to restore devastated stands or populations. The elevated need for immediate establishment of ex situ collections in forest trees as a safety net for climatic change is hampered by the absence of pertinent funding and long-term planning in the conservation of forest genetic resources. Progeny trials may have originally been established for breeding purposes, but they can be used to meet, at least in part, the above goal, as this study has shown.

## 4. Materials and Methods

### 4.1. Description of the Progeny Plantation

This study analyzed an *A. glutinosa* progeny plantation, established in 1998. Progeny plantation no. 09JBZ001 is located in the Dubrava Regional Branch of the State Forest Enterprise (SFE), Vytėnai forest district (55°3′57.4884″ N, 23°57′6.4908″ E) (Figure 5). The plantation contains 76 OP families, which originated from mother trees originally selected as plus trees for superior commercial traits in different stands (the selection traits for broadleaved plus trees are described in Verbylaitė [39]) that cover the whole of the natural distribution of the species in Lithuania. In total, 25 populations were used to select 1–5 plus trees. The seeds from the plus trees were collected in the wild populations, and later the originating seedlings were used to establish the Vytėnai progeny plantation. The purpose of the plantation was to ascertain which are the best performing OP families, and thus to determine which plus trees should be used to establish a clonal seed orchard. The plantation design is a randomized block (4 blocks), with 82–108 OP families per block and 10 plants per OP family repeat. The number of repeats per OP family is 1 to 6 (average 3.5). The planting distances are 1.5 m between plants in a row and 2 m between rows. At the time of sampling (2021), some of the initially planted trees were absent. The progeny plantation mortality rate is 42.6%.

### 4.2. Description of the GCUs

The black alder GCUs investigated in the present study are part of Lithuanian national forest seed base objects, which comply with minimum size requirements for dynamic in situ GCUs of forest tree genetic diversity [40,41]. Five (BAT, PAZ, SIM, SPA, and VIL) of the investigated GCUs are internationally recognized as such and are included in the EUFGIS information system database [42].

In Lithuania, there are two provenance regions for *A. glutinosa* that were delineated based on climatic regions, stand productivity, and phenological and morphological data analysis in accordance with species distribution in the country. In 2020, the delineation of provenance regions was specified based on molecular data.

### 4.3. Sampling

For this study in the Vytėnai progeny plantation, we sampled 22 OP families and 15–20 plants per OP family—420 plants in total (Appendix A, Figure 5). All OP families included in this study originated from different black alder populations (Appendix A, Figure 5, left). We selected OP families with no fewer than 15–20 surviving plants per family.

The genetic diversity of the progeny plantation was compared with that of 10 *A. glutinosa* GCUs (Appendix A, Figure 5, right). In 10 *A. glutinosa* GCUs (Appendix A, Figure 5, right), we sampled 58–61 plants each (597 plants in total). Half of the samples represented mature trees, the other half represented young regenerating trees. A detailed description of *A. glutinosa* in situ populations and sampling is given in Verbylaitė et al. [32]. In the Vytėnai progeny plantation, progenies from 5 out of 10 investigated GCUs were planted: JUO (OP families 135 and 139), MIK (OP family 97), PAZ (OP family 89), BAT (OP family 121), and VIL (OP family 103) (Figure 5). While sampling, care was taken to select only morphologically pure *A. glutinosa* trees, since in Lithuania, *A. glutinosa* and *Alnus incana* have an overlapping distribution [1,43] and the species are known to hybridize occasionally [44,45,46]. Identification of *A. glutinosa* samples followed the morphological description provided in Flora of Lithuanian TSR [47]. There are different methods described in the literature to identify naturally occurring hybrids, among them morphological leaf traits, specific Simple Sequence Repeat (SSR) markers, and chemotaxonomic markers [44,48,49]. In this case, sampled trees were selected based on leaf morphological traits as described in Jurkšienė et.al. [44]. After the SSR analysis, pure *A. glutinosa* status was confirmed [44].

When sampling progeny plantation and mature GCU trees, we collected cambium samples (using an electric drill at an up-to 3 cm drilling depth to avoid stem damage). The resulting sawdust was placed in 2 mL labeled Eppendorf tubes and kept at −20 °C until DNA extraction. The drill between the samples was sterilized using ethanol and flame to avoid sample cross-contamination. For the young regenerating trees, three or four leaf samples per tree were collected.

### 4.4. Molecular Analysis

The molecular analysis followed the methodology described in Verbylaitė et al. [32]. Prior to DNA extraction, 50–100 mg of plant material was homogenized in liquid nitrogen, and the resulting powder was immediately used for DNA extraction. DNA was isolated using the ATMAB extraction procedure [50]. We used 20 nuclear microsatellite markers for this study (Table 4). The microsatellite marker system for this study was chosen based on many desirable genetic attributes, including hypervariability, multiallelic nature, codominant inheritance, reproducibility, relative abundance, extensive genome coverage, chromosome specific location, and amenability to automation and high-throughput genotyping [51]. The PCR conditions, multiplexing, and amplification success rating were as described at Verbylaitė et al. [32]. Successfully amplified samples were sent for fragment size evaluation to Genoscreen Innovative Genomics (Lille, France), where an ABI 3730XL sequencer was utilized. To ensure the reproducibility of the SSR results, 10% of the samples were repeatedly amplified and independently evaluated. Microsatellite fragment sizes were scored using Geneious Prime^®^ version 2022.1.1. Scored microsatellite loci were checked for the presence of null alleles and scoring errors with MicroChecker software [36]. The main genetic diversity parameters were calculated using GenAlEx computer software (version 6.51b2) [52,53,54]. Allelic richness was calculated using FSTAT software [55]. The PCoA was conducted using GenAlEx computer software and a tri-distance matrix between families and populations as an input.

### 4.5. Sample Lots Used for Comparisons

For comparing genetic diversity parameters and allele numbers, we conducted analyses on different sample lots using allele frequency averages. The comparison scheme is presented in Figure 6. First, we compared single GCU populations that had their progenies in the progeny plantation (VIL, BAT, JUO, PAZ, and MIK) with the single OP progeny families (103, 121, 135, 139, 89, and 97) (comparison 1, Figure 6). Second, we compared single GCU populations with the six OP progeny families (89, 97, 103, 121, 135, and 139) grouped together (comparison 2, Figure 6) and with the total sample lot of 22 OP progeny families (comparison 3, Figure 6). Third, we compared five GCU populations (VIL, BAT, JUO, PAZ, and MIK) with that of six OP progeny families (89, 97, 103, 121, 135, and 139) originating from those populations (comparison 4, Figure 6), as well as with all 22 OP families (comparison 5, Figure 6). Fourth and finally, we compared all 10 GCU populations with all 22 OP families sample lot (comparison 6, Figure 6).

The next step in this study was to analyze the copy number of alleles present in the progeny plantation. This analysis helped determine how strong the Vytėnai progeny plantation is in the sense of ex situ collection robustness. The prediction of the allele number in the progeny plantation (76 OP families) for the different allele frequency categories was based on the best-fitted trendline (exponential, linear, logarithmic, polynomial, or power) for the average number of alleles in the different number of examined OP families (Figure 4).

## Figures and Tables

**Figure 1 plants-12-03986-f001:**
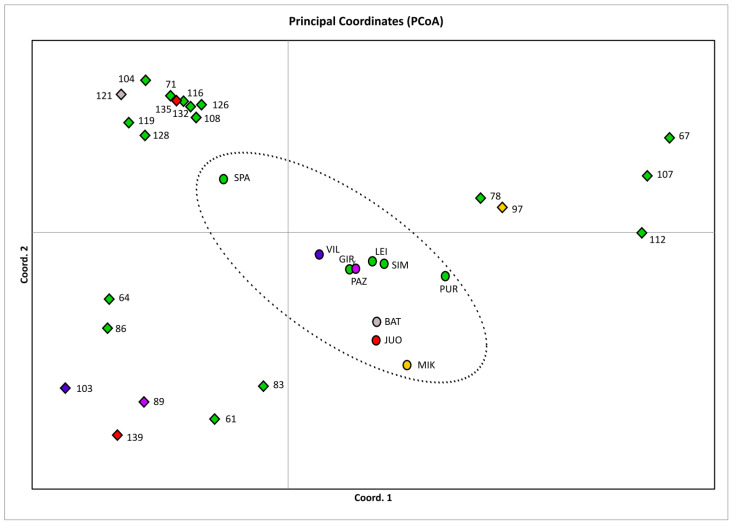
Principal coordinate analysis (PCoA) of the open pollinated (OP) families of the Vytėnai progeny plantation and of the wild GCU populations. The first two coordinates explain 33.39% of the total variation (the first coordinate explain 18.81% of the variation). The diamond-shaped dots show the OP families, with the family number next to it. The circular dots denote populations, with the abbreviated population name next to it. The gray, red, yellow, purple, and blue colors represent the population and families descended from those particular populations, while the color green represents all the other populations and families.

**Figure 2 plants-12-03986-f002:**
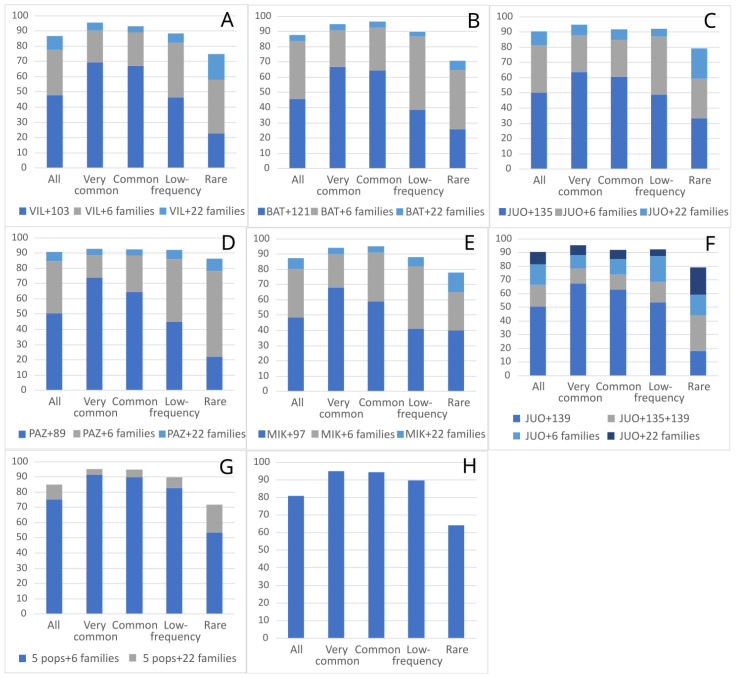
Diagrams showing proportion (%) of different category alleles found in different ex situ sample lots. (**A**–**F**)—one GCU population; (**G**)—five GCU populations (VIL, BAT, JUO, PAZ and MIK) combined; (**H**)—ten GCU populations combined.

**Figure 3 plants-12-03986-f003:**
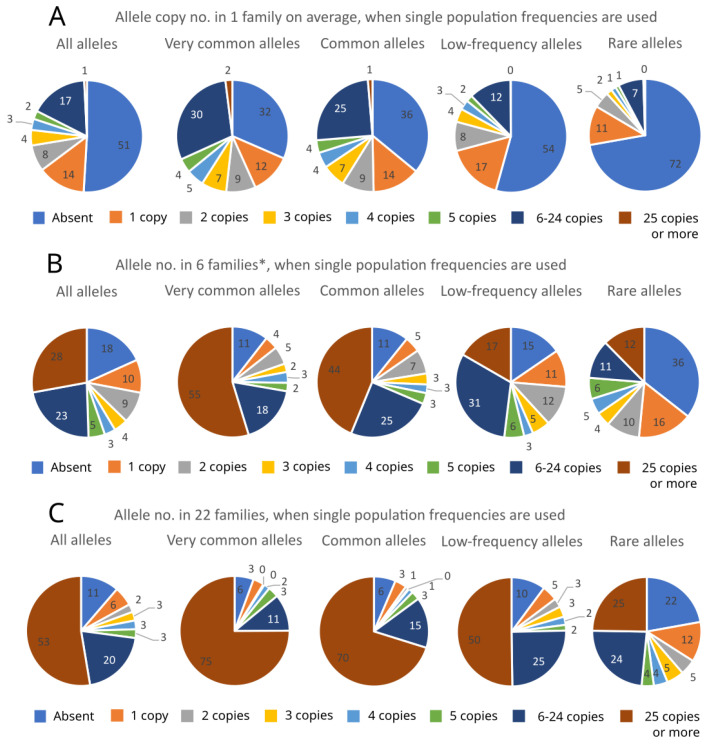
The proportion of alleles found in the *A. glutinosa* Vytėnai progeny plantation per different allele frequency categories represented for different sample groups ((**A**): allele copy number in one open pollinated (OP) family on average, when single population frequencies are used; (**B**): allele copy number in 6 OP families, when single population frequencies are used; (**C**): allele copy number in 22 OP families, when single population frequencies are used; (**D**): allele copy number in 6 OP families, when 5-population average frequencies are used; (**E**): allele copy number in 22 OP families, when 5-population average frequencies are used; (**F**): allele copy number in 22 OP families, when 10-population average frequencies are used). The graphs representing the actual allele numbers are given in Appendix A. * OP families no 89, 97,103, 121, 135, and 139.

**Figure 4 plants-12-03986-f004:**
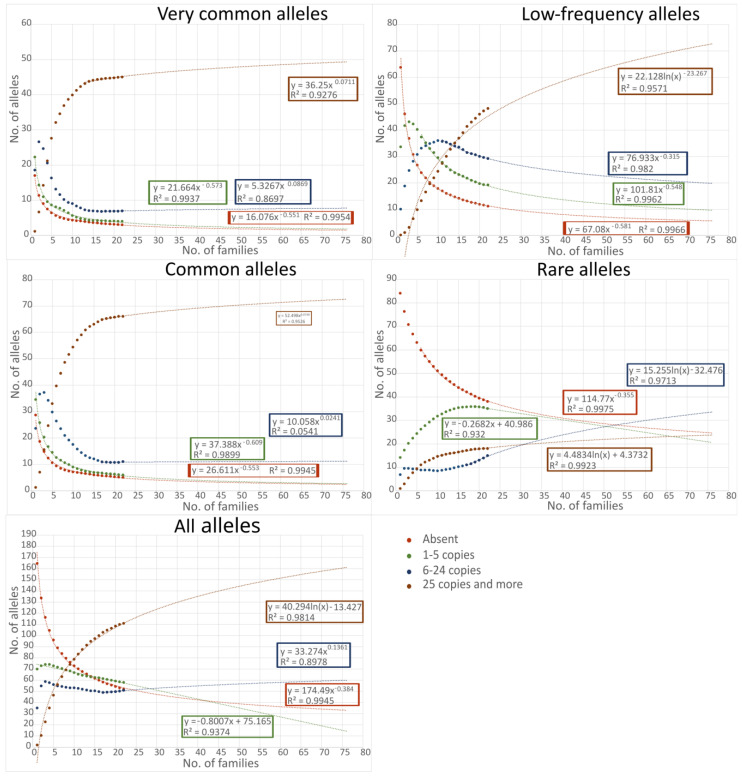
The change in the allele copy number in different allele categories (very common, common, low-frequency, and all) based on the number of open pollinated (OP) progeny families. The trendlines represent the expected copy number in each group if all progeny plantation OP families are included. The equations and R^2^ values describing the trendlines of each group are presented in corresponding color boxes next to each trendline.

**Figure 5 plants-12-03986-f005:**
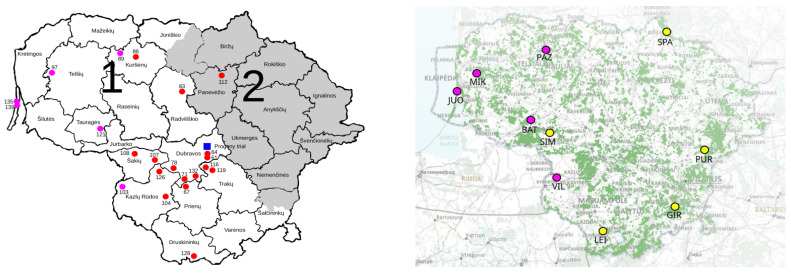
**Left**: Origin of *A. glutinosa* plus trees, of which seeds were used to establish progeny plantation (the mother tree no. = open pollinated family number next to its location). Mother trees from investigated GCU populations are marked in purple. Provenance regions are depicted in different background colors with numbers 1 and 2. The blue square shows the location of the progeny plantation. **Right**: *A. glutinosa* in situ GCU populations used for genetic diversity comparisons. The purple color depicts GCU populations represented in the progeny plantation. The green color on the map represents forests.

**Figure 6 plants-12-03986-f006:**
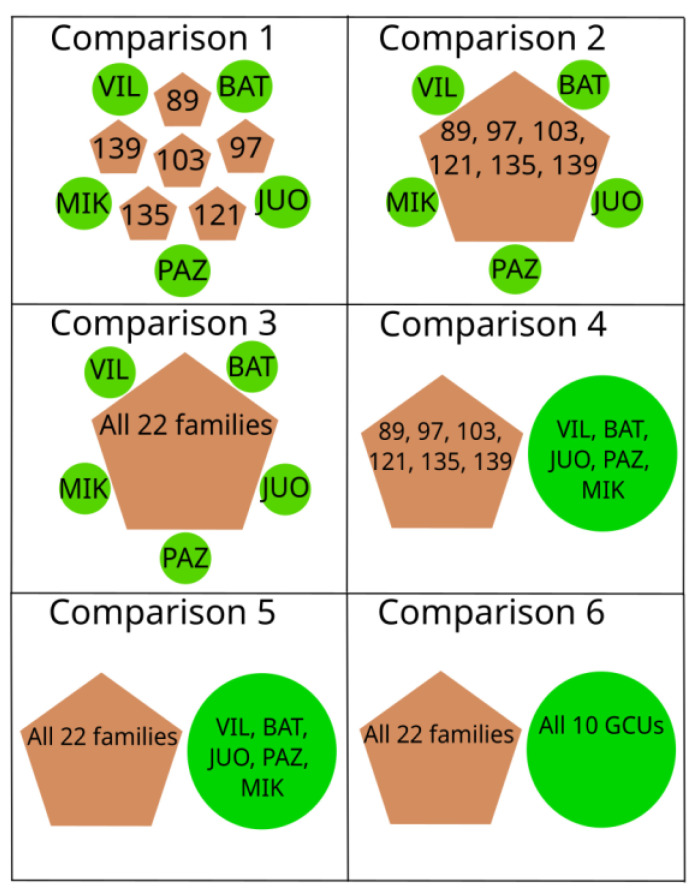
Scheme of different sample lots and the comparisons made among them. In each comparison, each brown pentagon is compared to each green circle.

**Table 1 plants-12-03986-t001:** The number of individuals to sample, as determined by the simulation study of Hoban [20].

		Low-Frequency Alleles	Local Alleles	All Alleles
Minimal collection (1 copy)	10–20 pops	50–70	100–200	200–800
	2–7 pops	40–60	30–80	150–550
Secure collection (5 copies)	10–20 pops	200–280	400–800	800–3200
	2–7 pops	160–320	120–320	600–2200
Robust collection (25 copies)	10–20 pops	800–1120	1600–3200	3200–12,800
	2–7 pops	640–960	480–1280	2400–8800

**Table 2 plants-12-03986-t002:** Genetic diversity parameters of *A. glutinosa* sample lots (GCUs and open pollinated (OP) families) compared in the present study. *Ne*—effective no of alleles; *Ho*—observed heterozygosity; *He*—expected heterozygosity; *F_IS_*—inbreeding coefficient; *Ar*—allelic richness, based on two diploid individuals. All genetic diversity parameters are provided with standard errors.

Sample Lot	*Ne*	*Ho*	*He*	*F_IS_*	*Ar*
BAT	5.08 ± 0.54	0.73 ± 0.04	0.75 ± 0.04	0.03 ± 0.03	2.89 ± 0.12
JUO	4.98 ± 0.46	0.64 ± 0.04	0.77 ± 0.02	0.18 ± 0.04	3.94 ± 0.08
MIK	4.46 ± 0.44	0.70 ± 0.04	0.72 ± 0.04	0.05 ± 0.04	2.79 ± 0.11
PAZ	4.60 ± 0.47	0.73 ± 0.04	0.73 ± 0.04	0.01 ± 0.03	2.81 ± 0.11
VIL	5.17 ± 0.49	0.76 ± 0.03	0.76 ± 0.03	0.01 ± 0.03	2.92 ± 0.11
5 * GCU pops combined	5.30 ± 0.51	0.71 ± 0.03	0.77 ± 0.03	0.08 ± 0.02	2.87 ± 0.05
10 GCU pops combined	4.97 ± 0.16	0.73 ± 0.01	0.75 ± 0.01	0.04 ± 0.02	2.88 ± 0.03
Family 89	3.38 ± 0.26	0.79 ± 0.06	0.67 ± 0.05	−0.17 ± 0.06	2.59 ± 0.10
Family 97	3.16 ± 0.30	0.70 ± 0.06	0.63 ± 0.04	−0.10 ± 0.07	2.49 ± 0.11
Family 103	3.44 ± 0.34	0.72 ± 0.06	0.64 ± 0.05	−0.10 ± 0.05	2.56 ± 0.13
Family 121	3.31 ± 0.22	0.85 ± 0.05	0.67 ± 0.03	−0.25 ± 0.04	2.59 ± 0.09
Family 135	3.88 ± 0.26	0.87 ± 0.03	0.72 ± 0.02	−0.19 ± 0.05	2.75 ± 0.08
Family 139	3.65 ± 0.28	0.82 ± 0.05	0.69 ± 0.04	−0.17 ± 0.05	2.67 ± 0.11
6 ** families combined	4.88 ± 0.42	0.79 ± 0.04	0.75 ± 0.03	−0.05 ± 0.02	2.61 ± 0.04
22 families combined	3.48 ± 0.06	0.77 ± 0.01	0.66 ± 0.01	−0.01 ± 0.02	2.61 ± 0.02

* JUO, VIL, BAT, PAZ, MIK populations. ** 89, 97, 103, 121, 135, 139 OP families.

**Table 3 plants-12-03986-t003:** The number of alleles found in different sample lots in natural GCU populations, divided into different categories.

Sample Lot	No. of All Alleles	No. of Very Common Alleles (>0.10)	No. of Common Alleles (>0.05)	No. of Low-Frequency Alleles (0.10 > 0.01)	No. of Rare Alleles (<0.01)
BAT	171	57 (33%)	90 (53%)	83 (49%)	31 (18%)
JUO	187	58 (31%)	89 (48%)	90 (48%)	39 (21%)
MIK	178	50 (28%)	88 (49%)	88 (49%)	40 (23%)
PAZ	168	61 (36%)	84 (50%)	71 (42%)	36 (21%)
VIL	189	59 (31%)	85 (45%)	82 (43%)	48 (25%)
5 pops combined	249	57 (23%)	87 (35%)	110 (44%)	82 (33%)
10 pops combined	272	59 (22%)	88 (32%)	107 (39%)	106 (39%)

**Table 4 plants-12-03986-t004:** Microsatellite markers used in this study. The Ag14, Ag20, and Ag23 (shaded) markers were excluded from calculations due to poor amplification. The number of alleles and the interval of alleles for these markers are based on GCU sample lot only.

Locus	Repeat Motif Size	No of Alleles	Allele Interval (In bp)	Primer Source References
A2	2	8	132–150	[56]
A22	2	15	151–181	[57]
A10	2	12	107–133	[58]
A35	2	16	216–248	[59]
A38	3	15	97–151	[59]
A7	2	13	168–200	[60]
A37	2	19	239–275	[61]
A26	2	19	341–377	[57]
Ag30	2	18	76–114	[62]
Ag05	2	17	136–172	[62]
Ag10	2	16	205–249	[62]
Ag14 *	2	31	271–341	[62]
Ag27	2	16	74–118	[62]
Ag35	2	24	164–214	[62]
Ag13	2	17	243–283	[62]
Ag25	2	24	73–131	[62]
Ag09	2	26	186–264	[62]
Ag20 *	2	19	283–333	[62]
Ag01	2	12	126–188	[62]
Ag23 *	2	9	336–370	[62]

* The Ag14, Ag20, and Ag23 markers were excluded from calculations due to poor amplification. The number of alleles and the interval of alleles for these markers are based on GCU sample lot only.

## Data Availability

The datasets generated and analysed during the current study are available from the corresponding author on reasonable request.

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
