# Peer review of "Can a Forest Tree Species Progeny Trial Serve as an Ex Situ Collection? A Case Study on Alnus glutinosa"

_plants, 2023, doi:10.3390/plants12233986_

Round 1

Reviewer 1 Report

Comments and Suggestions for Authors

My comments on the manuscript are as follow:

1.     I have read the manuscript, the information given are important and may of interest to a wider community and therefore worth reporting. The authors have carried out a progeny trial that consisted of several open pollinated families of Alnus glutinosa maintained for breeding purposes, and have looked for aspects if the same can also serve as an ex-situ conservation plantation by studying a case study of a Lithuanian progeny trial of Alnus glutinosa, a species of the riparian ecosystems that warrants priority conservation actions.

2.     The word families used in abstract and elsewhere is not correct and need changing. In plant sciences the term families should only mean/refer to the taxonomic unit and not the other way round to a group/cluster/association of the same individual/species. Although, the term population may be used but this is used for the number of trees sampled (Table 5).

3.     What is GCU, the term is used so many times i.e. form abstract till discussion but there are no details given till the final page of the MS in M&M section. GCU and all other abbreviations MUST be fully described at the place of their first mention, also what is the criteria/rationale to recognize any unit as a Genetic conservation Unit (GCU).

4.     P3-L102-103: how can this criteria be justified? or else the statement need rephrasing for clarity ……seeds sampled from 200-300 trees if it is a progeny collection, and at least 100 trees if it is a clonal collection will carry enough diversity, when the reverse/vice versa seems to carry more diversity.

5.     Table No. 1 (P2) provides little or no details of why this data is given, if it is to be used after all as a baseline for this study, there must be a judicious descriptions of that data set and its importance and only providing a reference source (Ref. 20) is not enough.

6.     Similarly, Figure 3 (P7) is adding little to the understanding of the alleles presence/variability, as almost all the sub-figures and especially “g & h” do not provide/add to the understanding of the results at all.

7.     Figure 4, is having important data but itself it is not conveying what it is supposed to do or portray. It is highly desirable to discuss this data within the MS. The legend is virtually the same for almost all sub-images (A-F) and thus of limited importance/use for the readers.

8.     The methodology section needs careful revision again for readers to understand, so that it is clear how many sample are used and what is/are the source/s of the sample/s.  At the moment, it is not clear at all how many samples/families/individuals have been studied. It is desirable to clarify the number of sample/individual used in the analyses at the start and find common grounds for introduction of these samples.

For example, P12 (L307 and onwards):

………….The trial contains 76 open pollinated families, which originated from mother trees and that cover the whole of the natural distribution of the species in Lithuania. Then details are given as……… In total, 25 populations were used to select 1-5 plus trees. ………the progeny trial also included 31 A. glutinosa open pollinated families, originating from Sweden. ……… again the authors have mentioned the trail design is a randomized block (4 blocks), with 82-108 families per block and 10 plants per family repeat……

Later in molecular analyses, the samples mentioned needs clarity (P13, L329 and onward):

………the progeny trial was compared to that of 10 A. glutinosa genetic conservation units (GCUs) (Table 5, Figure 6). In the progeny trial, progenies from five out of ten established GCUs were planted……..

After the SSR analysis the pure A. glutinosa status was confirmed (data not shown) [40], if this data is coming form Ref. 40, then data not shown is not needed. However, if it is done here the reference is not needed.

L345 ………….For sampling we have collected cambium samples from 22 open pollinated families (15-20 plants 346 per family, 420 plants in total).

Similarly, the SSRs related data is neither clear in results nor in discussion. Also critical details like the original source/s of SSRs and the sequences etc. are not indicated to and are of high priority. Hardly any mention of the DNA extraction, of amplification size, success rate of SSRs across the sample size etc. are not there.

9.     The samples/individual identification is mostly based on morphological variability and that's OK. But who is followed for these recordings/calculations of morphological attributes i.e. any descriptor used, citation of that may be provided. Neither I see who has authorised this study, and if any specimens of these morphotypes are submitted to any organization as voucher specimen/s?

At the moment details provided in M&M section are not clear enough and need stringent revisions/inputs.

10.  The results provided in the Results section in general are enough but it is highly desirable how reproducibility of the SSRs amplicons was ascertained. If anyone now is interested to repeat the same experiment, the current details are not enough in that case. Important to note, that the discussion section needs serious inputs and interpretation of the key results. How and why the method used/followed here is reliable enough for this or other similar studies in future. Until, this is justified in discussion or any other related section (may be results) the huge amount of data presented is not making the desired impact.

11.  Finally, there is plenty of data given/provided that cover the scope of the journal and scientific quality but the manuscript is structured in a way that is making it unclear as to what are the novel results and what is known earlier? Still the intext References referred to (Ref. No 20, 40) within the MS make it even more unclear.  

There are few minor issues and may be addressed:

a.     L92: authority for the species is provided yet, it is desirable to mention the authority at the start of the introduction section Alnus glutinosa (L.) Gaertn.

b.     There are typos and grammar related issues, specifically when elongated statements are used.

c.     Any abbreviations used needs to be described in full at their first place of mention.

d.     References needs to be re-assessed for consistently and followed as per the journal format/standards. All scientific names need to be in italic even if they are at the end in the references section.

Decision:

While the study is within the scope of the journal, and information may be handy and of wider interest. The MS in the current form cannot be accepted, it needs Revisions in a way to integrate the text in a more cohesive way to guide/enable readers to understand the key results aligned to the rationale of the study and its implications for similar studies in future.

Comments on the Quality of English Language

as given in the MS

Author Response

Dear reviewer, thank you for the valuable comments that helped us to improve the manuscript. The answers to each comment are given after each of the comments in blue.

Open Review

Quality of English Language

( ) I am not qualified to assess the quality of English in this paper
( ) English very difficult to understand/incomprehensible
( ) Extensive editing of English language required
( ) Moderate editing of English language required
(x) Minor editing of English language required
( ) English language fine. No issues detected

Yes

Can be improved

Must be improved

Not applicable

Does the introduction provide sufficient background and include all relevant references?

(x)

( )

( )

( )

Are all the cited references relevant to the research?

(x)

( )

( )

( )

Is the research design appropriate?

( )

(x)

( )

( )

Are the methods adequately described?

( )

(x)

( )

( )

Are the results clearly presented?

( )

(x)

( )

( )

Are the conclusions supported by the results?

(x)

( )

( )

( )

Comments and Suggestions for Authors

My comments on the manuscript are as follow:

  1. I have read the manuscript, the information given are important and may of interest to a wider community and therefore worth reporting. The authors have carried out a progeny trial that consisted of several open pollinated families of Alnus glutinosa maintained for breeding purposes, and have looked for aspects if the same can also serve as an ex-situ conservation plantation by studying a case study of a Lithuanian progeny trial of Alnus glutinosa, a species of the riparian ecosystems that warrants priority conservation actions.

  1. The word families used in abstract and elsewhere is not correct and need changing. In plant sciences the term families should only mean/refer to the taxonomic unit and not the other way round to a group/cluster/association of the same individual/species. Although, the term population may be used but this is used for the number of trees sampled (Table 5).

The word families in classic breeding experiments refers to full or half-sibs (progenies). The term is widely accepted and used (please see for example Eriksson G, Ekberg I, Clapham D. Genetics applied to forestry: an introduction. Department of Plant Biology and Forest Genetics, SLU; 2013.). To make it clearer we specified throughout the manuscript that it was open pollinated families.

  1. What is GCU, the term is used so many times i.e. form abstract till discussion but there are no details given till the final page of the MS in M&M section. GCU and all other abbreviations MUST be fully described at the place of their first mention, also what is the criteria/rationale to recognize any unit as a Genetic conservation Unit (GCU).

We have specified in the manuscript that GCU refers to the genetic conservation unit, and added the justification in materials and methods why the ten wild natural populations investigated in this study are referred as GCUs.

  1. P3-L102-103: how can this criteria be justified? or else the statement need rephrasing for clarity ……seeds sampled from 200-300 trees if it is a progeny collection, and at least 100 trees if it is a clonal collection will carry enough diversity, when the reverse/vice versa seems to carry more diversity.

The suggested clarification has been introduced to the main text.

  1. Table No. 1 (P2) provides little or no details of why this data is given, if it is to be used after all as a baseline for this study, there must be a judicious descriptions of that data set and its importance and only providing a reference source (Ref. 20) is not enough.

Table 1 was added to avoid lengthy and rather complicated reiteration of number of populations and plants needed to capture a desired portion of genetic diversity, as found by Hoban et al. 2019 in the simulations. The table has two references in the text, so there are some details. We feel, that this data gives background and helps the reader to understand better the data we present in our study.

  1. Similarly, Figure 3 (P7) is adding little to the understanding of the alleles presence/variability, as almost all the sub-figures and especially “g & h” do not provide/add to the understanding of the results at all.

Figure 3 demonstrates how well the wild meta-population/single populations are represented if only a portion of progeny trial is analyzed. The g and h presents the results of five/10 wild populations compared to 6/22 OP families. This gives the reader the perspective how the number of samples represent different portion of the genetic diversity. The a-f segments presents the single populations only. The idea behind demonstrating these data was to show how well the metapopulation is represented in the progeny trial. We have added the reasoning for using the metapopulation alleles and frequencies in the manuscript.

  1. Figure 4, is having important data but itself it is not conveying what it is supposed to do or portray. It is highly desirable to discuss this data within the MS. The legend is virtually the same for almost all sub-images (A-F) and thus of limited importance/use for the readers.

Figure 4 demonstrated the copy number of alleles, when figure 3 represents only proportion of presence/absence of alleles. It is a step forward to see how well in regards of ex-situ collection robustness the alleles represented in a progeny trial are. We have added the explanation in Materials and Methods section.

  1. The methodology section needs careful revision again for readers to understand, so that it is clear how many sample are used and what is/are the source/s of the sample/s.  At the moment, it is not clear at all how many samples/families/individuals have been studied. It is desirable to clarify the number of sample/individual used in the analyses at the start and find common grounds for introduction of these samples.

We have clarified the number of samples and their collection sites. We have rewritten some parts of the methodology, as well we have introduced some changes to Figure 6, in order to better present different sample lots. Additionally, the comparisons scheme has been drawn and added as figure 7 to the manuscript for clarity.

For example, P12 (L307 and onwards):

………….The trial contains 76 open pollinated families, which originated from mother trees and that cover the whole of the natural distribution of the species in Lithuania. Then details are given as……… In total, 25 populations were used to select 1-5 plus trees.

This part of the manuscript gives details what genetic material was used to establish a progeny trial. As the idea of the study was to see if the progeny trial can serve as exžsitu collection, we wanted to give the readers as explicit description of the progeny trial as possible.

………the progeny trial also included 31 A. glutinosa open pollinated families, originating from Sweden. ……… again the authors have mentioned the trail design is a randomized block (4 blocks), with 82-108 families per block and 10 plants per family repeat……

Later in molecular analyses, the samples mentioned needs clarity (P13, L329 and onward):

………the progeny trial was compared to that of 10 A. glutinosa genetic conservation units (GCUs) (Table 5, Figure 6). In the progeny trial, progenies from five out of ten established GCUs were planted……..

After the SSR analysis the pure A. glutinosa status was confirmed (data not shown) [40], if this data is coming form Ref. 40, then data not shown is not needed. However, if it is done here the reference is not needed.

L345 ………….For sampling we have collected cambium samples from 22 open pollinated families (15-20 plants 346 per family, 420 plants in total).

Similarly, the SSRs related data is neither clear in results nor in discussion. Also critical details like the original source/s of SSRs and the sequences etc. are not indicated to and are of high priority. Hardly any mention of the DNA extraction, of amplification size, success rate of SSRs across the sample size etc. are not there.

The detailed molecular analysis procedure is described in Verbylaitė et al. 2023. We have added a brief description of the procedures used in this manuscript as well. Also we have added a table with the markers used, where the source references for primers, repeat number, the number of alleles found, the interval of loci are presented.

  1. The samples/individual identification is mostly based on morphological variability and that's OK. But who is followed for these recordings/calculations of morphological attributes i.e. any descriptor used, citation of that may be provided. Neither I see who has authorised this study, and if any specimens of these morphotypes are submitted to any organization as voucher specimen/s?

We have added the citation of the descriptor used to identify Alnus glutinosa. Generaly, these types of research doesn’t require to collect voucher specimens therefore no specimens were collected and submitted to herbarium. The collected unused cambium and leaf material is being kept in -20oC temperature at Lithuanian Research Centre for Agriculture and Forestry. The study was funded by Research Council of Lithuania. Lithuanian Research Centre for Agriculture and Forestry (LAMMC) is a public body of Lithuania, and as such needs no authorization to conduct scientific research in state owned forests of Lithuania. Lithuanian State Forest Service (that is responsible for establishing progeny trials) and State Forest Enterprise which is the organization that is managing Forest Genetic Resources in Lithuania were informed about the research in the progeny trial, which for scientific research is deemed enough and requires no special authorization for LAMMC.

At the moment details provided in M&M section are not clear enough and need stringent revisions/inputs.

The materials and Methods section of the manuscript has been rewritten and clarified.

  1. The results provided in the Results section in general are enough but it is highly desirable how reproducibility of the SSRs amplicons was ascertained. If anyone now is interested to repeat the same experiment, the current details are not enough in that case. Important to note, that the discussion section needs serious inputs and interpretation of the key results. How and why the method used/followed here is reliable enough for this or other similar studies in future. Until, this is justified in discussion or any other related section (may be results) the huge amount of data presented is not making the desired impact.

The Materials and Methods section of the manuscript has been expanded and the requested information introduced. The explanation about the repeatability of the results added, as well as information about the marker of choice.

  1. Finally, there is plenty of data given/provided that cover the scope of the journal and scientific quality but the manuscript is structured in a way that is making it unclear as to what are the novel results and what is known earlier? Still the intext References referred to (Ref. No 20, 40) within the MS make it even more unclear.  

 The structure of the manuscript has been improved for better understanding.

There are few minor issues and may be addressed:

  1. L92: authority for the species is provided yet, it is desirable to mention the authority at the start of the introduction section Alnus glutinosa (L.) Gaertn.

Change introduced.

  1. There are typos and grammar related issues, specifically when elongated statements are used.

The English language and typos corrected by proofreading service.

  1. Any abbreviations used needs to be described in full at their first place of mention.

Changes introduced.

  1. References needs to be re-assessed for consistently and followed as per the journal format/standards. All scientific names need to be in italic even if they are at the end in the references section.

Changes introduced.

Decision:

While the study is within the scope of the journal, and information may be handy and of wider interest. The MS in the current form cannot be accepted, it needs Revisions in a way to integrate the text in a more cohesive way to guide/enable readers to understand the key results aligned to the rationale of the study and its implications for similar studies in future.

Comments on the Quality of English Language

as given in the MS

Submission Date

29 September 2023

Date of this review

19 Oct 2023 16:34:49

Reviewer 2 Report

Comments and Suggestions for Authors

[Minor] Line 58: please replace [12,13,14,15,16,17] with [12-17].

[Minor] Line 70: please replace (table 1; ) with (Table 1).

[Minor] Line 95: please replace [27,28,29,30,31,32] with [27-32].

[Minor] Line 141: Please remove SEs from the Table 2.

[Major] Line 143: Add to the graph of Figure 2 what is the percent explained by the Coord 1 and Coord 2.

[Major] Line 181: The quality of Fig. 3 is very low. Please replace it with higher quality ones.

[Major] Line 210: The quality of Fig. 4 is very low. Please replace it with higher quality ones.

[Major] Line 225: The quality of Fig. 3 is very low. Please replace it with higher quality ones.

[Minor] Line 272: delete the double "that".

[Minor] Line 339: 42-43

[Minor] Line 341: 44-45

[Minor] Line 348: please add the space between -20 and °C.

[Minor] Line 354: Table 4 should be in Supplementay materials.

[Minor] Line 360: Table 5 should be in Supplementay materials.

[Major] Line 362: A subsection on Material and Methods should be included with the protocol followed for the DNA extraction and Molecular Markers Analysis.

[Major] Line 363: An additional table with the main characteristics of 20 primers should be inserted.

Author Response

Dear reviewer, thank you for the valuable comments that helped us to improve the manuscript. The answers to each comment are given after each of the comments in blue.

Open Review

Quality of English Language

(x) I am not qualified to assess the quality of English in this paper
( ) English very difficult to understand/incomprehensible
( ) Extensive editing of English language required
( ) Moderate editing of English language required
( ) Minor editing of English language required
( ) English language fine. No issues detected

Yes

Can be improved

Must be improved

Not applicable

Does the introduction provide sufficient background and include all relevant references?

( )

(x)

( )

( )

Are all the cited references relevant to the research?

( )

(x)

( )

( )

Is the research design appropriate?

( )

(x)

( )

( )

Are the methods adequately described?

( )

( )

(x)

( )

Are the results clearly presented?

( )

( )

(x)

( )

Are the conclusions supported by the results?

( )

( )

(x)

( )

Comments and Suggestions for Authors

[Minor] Line 58: please replace [12,13,14,15,16,17] with [12-17].

Changed.

[Minor] Line 70: please replace (table 1; ) with (Table 1).

Changed.

[Minor] Line 95: please replace [27,28,29,30,31,32] with [27-32].

Changed.

[Minor] Line 141: Please remove SEs from the Table 2.

Changed.

[Major] Line 143: Add to the graph of Figure 2 what is the percent explained by the Coord 1 and Coord 2.

The explained percentage by different coordinates added.

[Major] Line 181: The quality of Fig. 3 is very low. Please replace it with higher quality ones.

The poor quality figure replaced with the better quality one. The excellent quality figure submitted to the journal as a separate pdf file.

[Major] Line 210: The quality of Fig. 4 is very low. Please replace it with higher quality ones.

The poor quality figure replaced with the better quality one. The excellent quality figure submitted to the journal as a separate pdf file.

[Major] Line 225: The quality of Fig. 3 is very low. Please replace it with higher quality ones.

The poor quality figure replaced with the better quality one. The excellent quality figure submitted to the journal as a separate pdf file.

[Minor] Line 272: delete the double "that".

Changed.

[Minor] Line 339: 42-43

Changed.

[Minor] Line 341: 44-45

Changed.

[Minor] Line 348: please add the space between -20 and °C.

Changed.

[Minor] Line 354: Table 4 should be in Supplementay materials.

Changed.

[Minor] Line 360: Table 5 should be in Supplementay materials.

Changed.

[Major] Line 362: A subsection on Material and Methods should be included with the protocol followed for the DNA extraction and Molecular Markers Analysis.

Materials and Methods section improved.

[Major] Line 363: An additional table with the main characteristics of 20 primers should be inserted.

Materials and Methods section improved, requested information inserted.

Submission Date

29 September 2023

Date of this review

16 Oct 2023 15:32:30

Reviewer 3 Report

Comments and Suggestions for Authors

I just reviewed the manuscript titled "Can a forest tree species progeny trial serve as an ex-situ collection? A case study in Alnus glutinosa" and here is my professional opinion about its quality. Although the manuscript represents a classic example of "salami-slicing" of the authors' previous article https://doi.org/10.3390/f14020330, it however contains a fair dose of scientifically-based information to be considered as a separate study. The main impression after reading the manuscript is its fuzziness. The authors should put additional effort to explain what are 5 populations, 10 populations, progeny trial, GCUs, families... In my opinion, a reader finds her/himself baffled with the terminology not clearly presented.

The second major problem is the resolution of Figures 3, 4 and 5 with literarily unreadable marks and the text therein. It must be improved.

The third major problem is the language usage. I mean, there are numerous punctuation errors and typos, but the main issue is in fact the syntax or the sentence structure and the text flow. To improve this, I suggest having the manuscript professionally edited by a native English speaker or a professional editing agency. It would significantly add the readability value and improve the manuscript’s overall informativeness.

The Discussion section is quite insubstantial and lacks information on similar studies dealing with the estimation of genetic variation of progeny trials for other tree species and the comparison with the presented results.

Specific remarks:

Main title: "on" instead of "in".

L12: Ljubljana instead of Ljubljan

L13: Bosnia instead of Bosnis

L17 and further in the text: terms "in situ" and "ex situ" are written without a hyphen

L18: however, "open-pollinated" contains a hyphen

L20-21: poor syntax in the second part of the sentence. Please rewrite to clarify.

L22: loci cannot be employed. Markers can instead. Genetic marker represents a DNA sequence variable enough to report on the variation on a specific locus in a population. A genetic locus is the position of a certain gene or a genetic marker on a chromosome.

L26: do not use abbreviations in Abstract unless being spelled out.

L27-28: "...harbors enough wild A. glutinosa populations" - vernacular expression. Please rewrite.

L29-30: too many information presented in a fuzzy way. Please clarify.

L45: compressions such as "haven't" are not used in scientific literature.

L70: shouldn't be "30" instead of "40" here?

L92: common plant names are not capitalized unless containing a proper noun (e.g., Lebanon cedar or Norway spruce)

L125-126: I wonder if the authors did not have access to other basic drawing tools to draw two circles of different colors, so had to employ an online tool and cite it as well?

L127-129: the content in the bracket is completely incomprehensible.

L135-136: this claim has no support in the PCoA biplot.

Table 2: I'm aware that GenAlEx in "HFP" sheet presents N as a number of individuals studied as a non-natural number (with decimals and with SE) owing the lack of information for a specific locus (loci) in a population, but this value is only for the calculation of observed heterozygosity, not to be present in a manuscript. Isn't it rather awkward studying 57.35 individuals in a population? Please remove the column. Moreover, how come mean N for 5 populations is 294.18 and for 10 populations is 59.36?

Table 2: is there a reason for the families to start with 103 and not with 61? Moreover, I suggest replacing "103 family" with "family 103" and so on.

Table 2: what 6 families represent?

Table 3: why 5 populations were extracted from the remaining ones?

L305: a study cannot analyze but present the results of the analysis...

L348: how the DNA was extracted? Please provide a brief protocol or a reference where it has been described.

L349: please note that ethanol cannot be used to sterilize a tool against DNA residues. It may be used for simple wipe off the plant material, but it is unable to degrade DNA. It is used for the DNA precipitation during the DNA isolation, isn't it?

L363: what platform was used for fragment size evaluation. If Sanger sequencing, please mention it.

Comments on the Quality of English Language

There are numerous punctuation errors and typos, but the main issue is in fact the syntax or the sentence structure and the text flow.

Author Response

Dear reviewer, thank you for the valuable comments that helped us to improve the manuscript. The answers to each comment are given after each of the comments in blue.

Open Review

Quality of English Language

( ) I am not qualified to assess the quality of English in this paper
( ) English very difficult to understand/incomprehensible
(x) Extensive editing of English language required
( ) Moderate editing of English language required
( ) Minor editing of English language required
( ) English language fine. No issues detected

Yes

Can be improved

Must be improved

Not applicable

Does the introduction provide sufficient background and include all relevant references?

( )

(x)

( )

( )

Are all the cited references relevant to the research?

( )

( )

(x)

( )

Is the research design appropriate?

(x)

( )

( )

( )

Are the methods adequately described?

( )

(x)

( )

( )

Are the results clearly presented?

( )

(x)

( )

( )

Are the conclusions supported by the results?

(x)

( )

( )

( )

Comments and Suggestions for Authors

I just reviewed the manuscript titled "Can a forest tree species progeny trial serve as an ex-situ collection? A case study in Alnus glutinosa" and here is my professional opinion about its quality. Although the manuscript represents a classic example of "salami-slicing" of the authors' previous article https://doi.org/10.3390/f14020330, it however contains a fair dose of scientifically-based information to be considered as a separate study. The main impression after reading the manuscript is its fuzziness. The authors should put additional effort to explain what are 5 populations, 10 populations, progeny trial, GCUs, families... In my opinion, a reader finds her/himself baffled with the terminology not clearly presented.

The materials and methods section has been improved. The separate descriptions for Vytėnai progeny trial, wild populations studied (aka. GCUs) described. Additionally the sampling effort presented better and comparisons scheme composed to better explain the sample lots that were compared.

The second major problem is the resolution of Figures 3, 4 and 5 with literarily unreadable marks and the text therein. It must be improved.

The better quality figures inserted in the manuscript. The excellent quality pdf figures submitted as separate files.

The third major problem is the language usage. I mean, there are numerous punctuation errors and typos, but the main issue is in fact the syntax or the sentence structure and the text flow. To improve this, I suggest having the manuscript professionally edited by a native English speaker or a professional editing agency. It would significantly add the readability value and improve the manuscript’s overall informativeness.

The manuscript after content editing sent for English proofreading services.

The Discussion section is quite insubstantial and lacks information on similar studies dealing with the estimation of genetic variation of progeny trials for other tree species and the comparison with the presented results.

Please note that this is a new approach to examined research problem. Evidently, to the authors’ knowledge, there are no similar publications in this respect. The potential comparison of comprehensive genetic diversity data of this progeny trial with other progeny trials would not give any particular answers or insight to the reader. Such diversity is always dependent on overall genetic diversity of the species, but more importantly on the particular genotypic composition of the mother trees which may differ a lot (i.e. in individual heterozygosity level). The Discussion aims at answering based on the Results one simple question: is the genetic diversity present in the progeny plantation enough to be considered as an ex situ conservation collection when compared to that the established in situ GCUs? The Discussion follows several line of evidence to clearly suggest in its conclusive statements that the answer is “yes”. Therefore it our thesis that the (revised) Discussion is not insubstantial as argued above.

Specific remarks:

Main title: "on" instead of "in".

Changed.

L12: Ljubljana instead of Ljubljan

Changed.

L13: Bosnia instead of Bosnis

Changed.

L17 and further in the text: terms "in situ" and "ex situ" are written without a hyphen

Changed.

L18: however, "open-pollinated" contains a hyphen

Changed.

L20-21: poor syntax in the second part of the sentence. Please rewrite to clarify.

English language editing service.

L22: loci cannot be employed. Markers can instead. Genetic marker represents a DNA sequence variable enough to report on the variation on a specific locus in a population. A genetic locus is the position of a certain gene or a genetic marker on a chromosome.

Changed.

L26: do not use abbreviations in Abstract unless being spelled out.

Abbreviations explained.

L27-28: "...harbors enough wild A. glutinosa populations" - vernacular expression. Please rewrite.

Rewritten.

L29-30: too many information presented in a fuzzy way. Please clarify.

Rewritten.

L45: compressions such as "haven't" are not used in scientific literature.

Changed.

L70: shouldn't be "30" instead of "40" here?

Corrected.

L92: common plant names are not capitalized unless containing a proper noun (e.g., Lebanon cedar or Norway spruce)

Corrected.

L125-126: I wonder if the authors did not have access to other basic drawing tools to draw two circles of different colors, so had to employ an online tool and cite it as well?

Yes, we have access to basic drawing tools. In this case it is just two circles, however, usually when doing Venn analysis, we have more different groups to consider. In this case we did the analysis in the usual way using an on-line tool. Diagram redrawn; citation omitted.

L127-129: the content in the bracket is completely incomprehensible.

The comparison scheme for different sample lots introduced in Materials and Methods and the reference to the scheme given.

L135-136: this claim has no support in the PCoA biplot.

The statement corrected.

Table 2: I'm aware that GenAlEx in "HFP" sheet presents N as a number of individuals studied as a non-natural number (with decimals and with SE) owing the lack of information for a specific locus (loci) in a population, but this value is only for the calculation of observed heterozygosity, not to be present in a manuscript. Isn't it rather awkward studying 57.35 individuals in a population? Please remove the column. Moreover, how come mean N for 5 populations is 294.18 and for 10 populations is 59.36?

The column removed from the table. The mean number for 10 populations were wrongly presented. It should have been 593.36.

Table 2: is there a reason for the families to start with 103 and not with 61? Moreover, I suggest replacing "103 family" with "family 103" and so on.

Corrected.

Table 2: what 6 families represent?

These 6 OP families are direct progenies of five of the populations investigated. We tried to clarify this issue in materials and methods section.

Table 3: why 5 populations were extracted from the remaining ones?

Those 5 populations had progenies in the Vytėnai progeny trial. That is the reason for extracting these five. We tried to clarify this issue in materials and methods section.

L305: a study cannot analyze but present the results of the analysis...

Corrected.

L348: how the DNA was extracted? Please provide a brief protocol or a reference where it has been described.

Details for DNA extraction provided.

L349: please note that ethanol cannot be used to sterilize a tool against DNA residues. It may be used for simple wipe off the plant material, but it is unable to degrade DNA. It is used for the DNA precipitation during the DNA isolation, isn't it?

The statement corrected. We have used ethanol to wash away the residual plant material and then we have used flame to remove residual ethanol, which also sterilizes the tip of the drill as flame degrades DNA.

L363: what platform was used for fragment size evaluation. If Sanger sequencing, please mention it.

The information of Genetic Analyzer used for fragment size evaluation added to the materials and methods.

Comments on the Quality of English Language

There are numerous punctuation errors and typos, but the main issue is in fact the syntax or the sentence structure and the text flow.

The English editing service employed.

Submission Date

29 September 2023

Date of this review

16 Oct 2023 19:24:33

Reviewer 4 Report

Comments and Suggestions for Authors

Review of the mss. 'Can a forest tree species progeny trial serve as an ex-situ collection? A case study in Alnus glutinosa'
by
Rita Verbylaitė, Filippos A. Aravanopoulos, Virgilijus Baliuckas, Aušra Juškauskaitė and Dalibor Ballian

1. Please, explain the 'GCU' abbreviation in the 'Results' section at the first mentioning, because the 'Material and methods' section where it is explained is in the end of the text.
2. line 118: use a modal verb in 'while in natural populations 272'
3. lines 118-119: 'In the GCUs 52 alleles found, were absent in the ex-situ sample lot, while 15 alleles found in the trial, were absent in the in-situ samples' - consider grammatical revising.
4. Table 2 - move to Supplementary Materials. Explain the choice of populations and families highlighted in bold. Provide only mean and the most contrasting values of some necessary for explanation parameters in the text.
5. Fig. 2: Explain symbol shapes and colors, as well as letters and numbers at the symbols. What data exactly was used for the PCoA? Was that a binary table of allele presence/absence in individual trees or these data were lumped for each family/population?
6. Fig. 3 is difficult to comprehend. Why the samples 'a' to 'h' were organized in such combinations? What were the reasons for that?
7. Fig. 4 is even more difficult to comprehend. Letters and figures at each of the pie charts are too small to be read clearly.
8. Fig. 5 is unclear. The letter size at diagram axes and inside colored boxes makes them completely unreadable. Enlarge font size in figs. 3 to 5.
9. Instead of calculating allele category proportions in variously sampled lots I suggest a Bayesian analysis of wild populations vs. the artificial trial using the STRUCTURE program. This may show the number of differentiated genetic pools in the whole sample and whether specimens from the trial are representative of these genetic pools without numerous hardly comprehensible calculations provided by the authors.
10. The authors use 'inbreeding coefficient' F but do not indicate how it was calculated. Was it Fit, Fis or Fst coefficient? Or something else appropriate to analyse multiple loci data?
11. Tables 4 and 5 should be moved to supplementary materials.

General conclusion:

The authors should re-analyse their data using more appropriate sampling design, e.g. comparing all wild populations vs. all trial families; all trial families vs. only those wild populations from which the trial material originated; only trial material to check for admixture if these plants are in the generative state and may represent plants of the second generation originated from crosses within the trial. The latter issues are not clear from the manuscript. The sampling desing is obscure. Please, indicate clearly what geographical coordinates in the table 4 mean? Are they coordinates of separate plantations with trees originating from certain plus-trees sampled from wild populations? Please, provide a detailed explanation what the terms 'family' and 'open pollinated family' mean in the context of your study. What was the age of the trees from the trial and were they all adult trees in the generative state or not? That was the age of the trees sampled in the wild? Beside this, please, provide information about those plants originated from Sweden. Were they also sampled or not? I think that the term 'trial' (evidently reflecting the purpose of establishin the plantation of Alnus glutinosa) should be replaced by the more clear term 'plantation' or 'plantations' if the authors' speak of multiple artificial stands separated from each other by a considerable distance. The latter is not evident from the text.

So, the study needs to be re-designed and the data re-analysed using appropriate methods to reveal the genetic structure (if any exists) within and between the sampled wild populations and artificial stands. Without that have been done all the authors' calculations of SSR allele frequencies are not convincing.

Comments on the Quality of English Language

The English is readable, however, moderate grammatical improvements, especially as to the usage of articles and modal verbs, should be done.

Author Response

Dear reviewer, thank you for the valuable comments that helped us to improve the manuscript. The answers to each comment are given after each of the comments in blue.

Open Review

Quality of English Language

( ) I am not qualified to assess the quality of English in this paper
( ) English very difficult to understand/incomprehensible
( ) Extensive editing of English language required
(x) Moderate editing of English language required
( ) Minor editing of English language required
( ) English language fine. No issues detected

Yes

Can be improved

Must be improved

Not applicable

Does the introduction provide sufficient background and include all relevant references?

( )

(x)

( )

( )

Are all the cited references relevant to the research?

(x)

( )

( )

( )

Is the research design appropriate?

( )

( )

(x)

( )

Are the methods adequately described?

( )

( )

(x)

( )

Are the results clearly presented?

( )

( )

(x)

( )

Are the conclusions supported by the results?

( )

( )

(x)

( )

Comments and Suggestions for Authors

Review of the mss. 'Can a forest tree species progeny trial serve as an ex-situ collection? A case study in Alnus glutinosa'
by
Rita Verbylaitė, Filippos A. Aravanopoulos, Virgilijus Baliuckas, Aušra Juškauskaitė and Dalibor Ballian

1. Please, explain the 'GCU' abbreviation in the 'Results' section at the first mentioning, because the 'Material and methods' section where it is explained is in the end of the text.

The GCU abbreviation explained.

  1. line 118: use a modal verb in 'while in natural populations 272'

English editing services employed.

  1. lines 118-119: 'In the GCUs 52 alleles found, were absent in the ex-situ sample lot, while 15 alleles found in the trial, were absent in the in-situ samples' - consider grammatical revising.

English editing services employed.

  1. Table 2 - move to Supplementary Materials. Explain the choice of populations and families highlighted in bold. Provide only mean and the most contrasting values of some necessary for explanation parameters in the text.

The explanation of sample lots and comparisons made in the Materials and Methods section. We have added the scheme for comparisons as figure 7. Table 2 in the main manuscript body was made smaller, keeping only the most important values.

  1. Fig. 2: Explain symbol shapes and colors, as well as letters and numbers at the symbols. What data exactly was used for the PCoA? Was that a binary table of allele presence/absence in in the dividual trees or these data were lumped for each family/population?

The symbols and colors explained in the figure legend. The input data and software used to generate PCoA added to the Materials and Methods section.

  1. Fig. 3 is difficult to comprehend. Why the samples 'a' to 'h' were organized in such combinations? What were the reasons for that?

The scheme, showing comparisons added to the manuscript as figure no 7. The reason for comparing the sample lots in this way was to show how well single population genetic diversity is preserved if single OP family (15-20 seeds), descended from the population is sampled, 6 families (approx.. 100 trees) are sampled, and 22 families (all 420 investigated trees) are sampled. The g and h already represents the multipopulation that is present in Lithuania for black alder. In g, five population average are analyzed, and in h – 10 population average.

The reasoning was explained in the manuscript.

  1. Fig. 4 is even more difficult to comprehend. Letters and figures at each of the pie charts are too small to be read clearly.

Better quality figure inserted.

  1. Fig. 5 is unclear. The letter size at diagram axes and inside colored boxes makes them completely unreadable. Enlarge font size in figs. 3 to 5.

Figure fonts enlarged and better quality figures inserted in the manuscript file. The excellent quality figures submitted to journal as separate files.

  1. Instead of calculating allele category proportions in variously sampled lots I suggest a Bayesian analysis of wild populations vs. the artificial trial using the STRUCTURE program. This may show the number of differentiated genetic pools in the whole sample and whether specimens from the trial are representative of these genetic pools without numerous hardly comprehensible calculations provided by the authors.

Prior to doing allele calculations we made the structure analysis. The Bayesian structuring revealed that the best fit is when K2 is used. This type of structuring does not reveal to us all hidden allelic diversity and is unable to answer the questions we were trying to answer with more complicated calculations.

K2

K14

K32

Of course you can see that the plants within families are much more similar than the wild populations, but how well these populations are represented in the progeny trial genetically is difficult to say only from the Bayesian clustering.

  1. The authors use 'inbreeding coefficient' F but do not indicate how it was calculated. Was it Fit, Fis or Fst coefficient? Or something else appropriate to analyse multiple loci data?

The given inbreeding coefficient was replaced with Fis estimate, calculated with FSTAT software.

  1. Tables 4 and 5 should be moved to supplementary materials.

Tables 4 and 5 moved to supplementary materials.

General conclusion:

The authors should re-analyse their data using more appropriate sampling design, e.g. comparing all wild populations vs. all trial families;

The mentioned analysis was already presented in the manuscript -  Figure 3 h, figure 4 f.

 all trial families vs. only those wild populations from which the trial material originated;

The mentioned analysis was already presented in the manuscript - Figure 3 g, figure 4 e.

only trial material to check for admixture if these plants are in the generative state and may represent plants of the second generation originated from crosses within the trial.

We have clarified in the manuscript that we have sampled the progenies of the mother plus trees, that grow in the wild populations. It was not our intention to study the second generation. To do so we need to conduct a separate study. As the progeny trial has a clear scheme and it is maintained to remove any black alder seedlings that emerge, we are positively sure that we have collected the progenies. The generative state of the sampled material was not noted as it was beyond the scope of the study.

The latter issues are not clear from the manuscript. The sampling desing is obscure. Please indicate clearly what geographical coordinates in the table 4 mean? Are they coordinates of separate plantations with trees originating from certain plus-trees sampled from wild populations? Please, provide a detailed explanation what the terms 'family' and 'open pollinated family' mean in the context of your study. What was the age of the trees from the trial and were they all adult trees in the generative state or not? That was the age of the trees sampled in the wild?

The coordinates given in Table 4 are the coordinates of the mother trees, from which the seeds were collected to establish Vytėnai progeny trial. The collected seeds were sown in the nursery, and later 2 year old seedlings were planted to establish Vytėnai progeny trial. So the progenies are direct first generation descendants from the wild population mother trees (plus trees selected in the wild). The age of the mother trees (plus trees in the wild populations) at the time of sampling varied, but all trees were mature approximately 50-60 years of age. The age of the progenies at the time we did the progeny trial sampling was 25 years.

Open pollinated family in the case of our study means that the source of the seeds were wild plus trees in the wild populations. The seeds from the same tree are half-sibs (open pollinated family), as the plus tree was freely pollinated by the pollen of the wild population of the plus tree.

Beside this, please, provide information about those plants originated from Sweden. Were they also sampled or not? I think that the term 'trial' (evidently reflecting the purpose of establishin the plantation of Alnus glutinosa) should be replaced by the more clear term 'plantation' or 'plantations' if the authors' speak of multiple artificial stands separated from each other by a considerable distance. The latter is not evident from the text.

The open pollinated families from Sweeden were mentioned in the manuscript only to describe the progeny trial. These plants were not sampled, but as we sampled not the seeds but the trees of the progeny trial, the Swedish descent trees have no impact on this study. The sentence describing the Sweden descent progenies in the plantation removed from the manuscript.

The term trial replaced with the term plantation.

So, the study needs to be re-designed and the data re-analysed using appropriate methods to reveal the genetic structure (if any exists) within and between the sampled wild populations and artificial stands. Without that have been done all the authors' calculations of SSR allele frequencies are not convincing.

Comments on the Quality of English Language

The English is readable, however, moderate grammatical improvements, especially as to the usage of articles and modal verbs, should be done.

The English proofreading service employed.

Submission Date

29 September 2023

Date of this review

14 Oct 2023 11:53:33

Round 2

Reviewer 1 Report

Comments and Suggestions for Authors

Dear Editors, 

I have been through the MS file (V2) and the revised version is very much improved and it incorporates all what I wished to see. So it have no issues to recommend it for publication. 

Reviewer 3 Report

Comments and Suggestions for Authors

I find all reviewers' concerns properly addressed. I suggest proceeding to publication.

Comments on the Quality of English Language

Several typographic errors.

Reviewer 4 Report

Comments and Suggestions for Authors

The manuscript is very much improved compared to its original version.The  English is clear. The only one issue I suggest to change is to exclude the Figure 1. I think the Figure 1 is uninformative and redundant, the same information contains in the text (lines 260-265).